# Study on static properties and mechanism of basalt fibre reinforced cement cured red sandstone soil

Yao Long[1], Jun-Hua Chen [ID][2]*, Jie-Jie Mu[2], Qi-Yun Wang[3]

1 Hunan Provincial Engineering Research Center for Intelligent Operation and Maintenance of High-Speed Rail and Urban Rail Transit, Hengyang, China, 2 School of Architecture and Transportation Engineering, Guilin University of Electronic Technology, Guilin, China, 3 College of Civil and Construction Engineering, Hunan Institute of Technology, Hengyang, China

* jhchan@126.com

## Abstract

To enhance the suitability of red sandstone as a railway roadbed fill, basalt fiber (BF) was utilized to modify cement cured red sandstone soil. The study commenced with the determination of the optimal cement admixture in improved red sandstone soil through disintegration testing. Following this, unconfined compressive strength (UCS) tests, undrained and unconsolidated shear (UU) tests were conducted to assess the impact of BF on the strength and deformation characteristics of the cement cured red sandstone soil. Finally, the intrinsic and damage mechanisms through which BF improves the mechanical properties of cement cured red sandstone soil were elucidated in conjunction with scanning electron microscopy (SEM) testing. The results of the study indicate that cement significantly enhances the water stability of red sandstone soil. The disintegration of the specimens effectively ceased once the cement dosage exceeded 4%. The addition of BF significantly enhances the strength of cement cured red sandstone soil. As the BF content increases, the UCS and peak deviatoric stress exhibit an initial increase followed by a decrease. At the optimal BF dosage of 6‰, the UCS improved 24.48%～25.40%, while the peak deviatoric stress improved 31.13%～39.48%. The incorporation of BF also enhanced the deformation and stability properties of the cement cured red sandstone soil, resulting in increased elastic modulus and failure strain. However, the soil brittleness index exhibited varying degrees of reduction, while ductility was improved. The SEM test results indicate that cement primarily provides cohesion between soil particle. BF effectively inhibits the generation and propagation of cracks through the adhesive properties of cement and its interfacial friction with soil particle, as well as by forming a three-dimensional reinforcing network. The research result demonstrates that the use of BF to enhance cement-cured red sandstone soil significantly improves its mechanical properties, offering a sustainable method for strengthening railway foundations and contributing to advancements in civil engineering applications.

**Data availability statement:** All relevant data are within the paper and its Supporting Information files.

**Funding:** (1) Hunan Natural Science Foundation Sectoral Joint Fund (2024JJ8021). (2) Hunan Provincial Department of Education Scientific Research Project (22B0958). (3) Guangxi Natural Science Foundation Project (2022GXNSFAA035485). (4) Science and Technol-ogy Base and Talent Special Project (GUIKE AD21220051). (5) Scientific Research Pro-ject of the Hunan Provincial Department of Education, China (21B0803). The contributor role statement is as follows: Hunan Provincial Natural Science Foundation (No. 2024JJ8021) Conceptualization; Resources; Writing—review and editing; Project administration Scientific Research Project Fund of Hunan Provincial Department of Education (No. 22B0958) Conceptualization; Resources; Writing—review and editing; Project administration Scientific Research Project of the Hunan Provincial Department of Education, China (No. 21B0803) Conceptualization; Resources; Writing—review and editing; Project administration.

**Competing interests:** The authors have declared that no competing interests exist.

## 1. Introduction

Red sandstone is a collective term for sedimentary rock such as mudstone, sandy mudstone, muddy fine sandstone, sandstone, and sandy shale that are rich in iron oxides [1,2]. The strength of these rocks varies significantly due to differences in mineral composition and cementing materials [3]. Under the influence of atmospheric condition or dry-wet cycle, red sandstone is prone to disintegration and crumbling, leading to reduced strength and poor engineering properties. Consequently, it is often used as a disposal material in roadbed construction [4]. However, red sandstone is widely distributed in China. In order to reduce costs and protect the environment, it is necessary to improve red sandstone before using it as roadbed fill [5,6].

The primary cause of failure in red sandstone roadbed is the water-induced disintegration and softening characteristics of red sandstone [7]. The roadbed is prone to subsidence when exposed to water, resulting in undesirable engineering issues such as pavement cracking [8]. Therefore, when improving red sandstone roadbed fill, it is essential to consider not only the strength and bearing capacity, but also the disintegration properties [9]. In order to enhance the poor engineering characteristics of red sandstone roadbed fill in practical applications, pre-disintegration through natural or artificial mechanical action is commonly employed to mitigate the water activity of red sandstone [10]. Simultaneously, the addition of external additives is used to improve the road performance of the red sandstone filler and enhance the stability of the roadbed [11]. Many scholars have conducted experimental studies on the improvement of red sandstone roadbed fill. They found that cement or lime is more effective as a modifier, significantly enhancing the strength and disintegration resistance of the fill [12]. However, soil improved with cement, lime, and similar materials can lead to several issues, such as susceptibility to brittle failure under loading [13–14]. Fibers are widely used as a reinforcing material for soil improvement. Many scholars have found that fiber is not only effective in enhancing the brittle damage characteristics of soil but also in improving their strength. Sujatha et al. [15] studied the mechanical properties of glass fiber reinforced soil. The results indicated that the glass fiber enhanced the ductility of the soil. Unreinforced soil exhibit distinct shear surfaces, whereas reinforced soil display multiple shear breaks and bulges, accompanied by a network of small cracks connected by fiber. Rashid et al. [16] investigated the bearing properties of red hemp fiber geotextiles placed on and within a sand layer through physical model tests. The study concluded that the use of red hemp fiber geotextile can increase the bearing capacity of sandy soil by up to 414.9%. S. Reehana et al. [17] investigated the mechanical response of fiber reinforced expansive soil under cyclic loading. The results indicated that the inclusion of fiber improved the dynamic shear modulus and damping ratio while reducing permanent strain as circumferential pressure increased. Consequently, some scholars have sought to enhance the deformation characteristics of cured soil through fiber reinforcement. António A. S. et al. [18] investigated the strength of steel fiber reinforced stabilized soil through tests such as UCS test, Split Strength (STS) test, and Flexural Strength (FS) test. The addition of steel fiber was

found to reduce the brittleness of binder-stabilized soil. M. Olgun. [19] conducted non-shrinkage compressive strength and split tensile strength tests to assess the effect of polypropylene fiber on the mechanical properties of chemically stabilized clayey soil using cement and fly ash. The results indicate that the addition of fiber to the stabilized soil resulted in a significant increase in both compressive strength and tensile strength values. The highest strength values were obtained with 12 mm fiber at a fiber content of 0.5% to 0.75%.

Current research on the improvement of roadbed soil primarily focuses on clay and expansive soil. The evaluation of improvement effects centers on factors such as expansion, load-bearing ratio, and UCS [20]. Regarding red sandstone, a unique type of roadbed fill, the goal of improvement is not only to enhance its strength and bearing capacity, but also to eliminate its disintegrability and improve its water stability simultaneously. Therefore, this paper presents an indoor improvement tests study conducted on common red sandstone filler from the Hunan region, utilizing cement as the curing agent and BF as the reinforcing material. First, the water stability of cement cured red sandstone soil was assessed by selecting the optimal cement admixture through disintegration tests. Secondly, following the determination of the optimal cement content, BF of varying dosages were incorporated into the cement cured red sandstone soil. The effects of the BF on the strength and deformation characteristics of the soil were analyzed through UCS tests and UU tests. And the improvement effect was evaluated based on these test results. Finally, the improvement and damage mechanisms of cement composite BF improved red sandstone soil were analyzed through SEM tests.

## 2. Test Materials and methods

### 2.1. Test materials

The red sandstone studied in this paper was sourced from Hengyang, Hunan Province. This red sandstone disintegrates into blocks after 24 hours of water immersion. In order to reduce the water activity of the red sandstone, it was crushed using machinery. Subsequently, the soil that passed through a 2 mm sieve was selected for testing. The basic physical properties of this soil sample were measured according to the Code for soil test of railway engineering (TB 10102–2023) [21], and the results are shown in Tables 1 and 2. According to the Code for Design of Railway Earth Structure (TB 10001−2016) [22], this red sandstone soil is classified as well graded fine sand (SW). And it can be used as the sub-base of railway roadbed and embankment body fill after improvement.

Cement is selected as P.O 42.5 grade ordinary silicate cement, and its technical parameters are shown in Table 3.

**Table 1. Basic physical properties of red sandstone soil.**

| Maximum dry density(g·cm³) | Optimum moisture content(%) | Specific gravity | Liquid limit(%) | Plastic limit(%) | Plasticity index |
|---|---|---|---|---|---|
| 2.04 | 11.46 | 2.72 | 25.01 | 11.11 | 13.90 |

**Table 2. Particle composition of red sandstone soil.**

| Diameter of hole (mm) | 2 | 1 | 0.5 | 0.25 | 0.074 | Coefficient of nonuniformity | Coefficient of curvature |
|---|---|---|---|---|---|---|---|
| Percentage of soil particle less than the pore size/ (%) | 100 | 89.6 | 79.96 | 39.06 | 14.3 | 5.8 | 1.17 |

**Table 3. Technical parameters of cement.**

| Test projects | Loss on ignition(%) | Specific surface area(m²·kg⁻¹) | Setting time (min) | | UCS(MPa) | |
|---|---|---|---|---|---|---|
| | | | initial setting | final setting | 3d | 28d |
| Test results | 3.5 | 360 | 175 | 235 | 27.5 | 49 |
| Technical requirement | ≤5.0 | ≥300 | ≥45 | ≤600 | ≥23.0 | ≥42.5 |

   

Basalt fibre is a new type of green material [23,24]. Higher strength and modulus of elasticity, along with excellent corrosion resistance and durability, are the primary factors contributing to its superior performance compared to chemical and plant-based fibre [25]. BF's main performance indicators are shown in Table 4.

## 2.2. Test design

Based on common dosage ranges for cement cured soil, the proposed cement dosages (mass percentage of red sandstone soil) are 3.0%, 4.0%, and 5.0% [22]. Considering the economics of BF, the proposed BF dosages (mass percentage of red sandstone soil) are 2‰, 4‰, 6‰, and 8‰. For ease of presentation, each proportion is represented by a simple code, such as C4BF6, which indicates a cement dosage of 4% and a BF dosage of 6‰. This coding system applies to all the ratios. The compaction test was conducted to determine the maximum dry density and optimal moisture content for each mixing ratio. The specific proportions of test specimens are presented in Table 5. In order to evaluate the effect of BF on the mechanical properties of cement cured red sandstone soil, the optimum cement proportion was first determined through disintegration tests. Subsequently, UCS tests and UU tests were conducted on specimens of both cement cured red sandstone soil (with the optimum proportion) and different BF reinforced cement cured red sandstone soil.

## 2.3. Test methods

The test flow chart for this study is presented in Fig 1. The testing procedure was conducted in accordance with the Code for soil test of railway engineering (TB 10102−2023) [21].

First, each air-dried soil sample is spread evenly in the tray, and the required amount of water is uniformly sprayed onto the test specimen. The soil is then thoroughly mixed with a mixing tool and placed in a plastic bag for soaking. After 4 hours of immersion, the cement is weighed according to the pre-calculated mass and added to the specimen. An additional 3%

**Table 4. Main performance indicators of BF.**

| Fibre length(mm) | Monofilament diameter(μm) | Density(g·cm³) | Elastic modulus(GPa) | Tensile strength(GPa) | Ultimate tensile ratio(%) |
|---|---|---|---|---|---|
| 12 | 11 | 2.64 | 91~110 | 3~4.8 | 2.6~3.1 |

**Table 5. Proportions of test specimens.**

| Test type | Specimen number | Cement(%) | BF(‰) | Maximum dry density (g·cm⁻³) | Optimum moisture content(%) | Degree of compaction(%) |
|---|---|---|---|---|---|---|
| Disintegration test | C0BF0 | 0 | 0 | 2.02 | 12.58 | 95 |
| | C3BF0 | 3 | | 1.98 | 12.99 | |
| | C4BF0 | 4 | | 1.96 | 13.69 | |
| | C5BF0 | 5 | | 1.95 | 14.16 | |
| UCS test | C4BF0 | 4 | 0 | 1.96 | 13.69 | 90, 92, 93, 95 |
| | C4BF2 | | 2 | 1.94 | 13.70 | |
| | C4BF4 | | 4 | 1.93 | 13.69 | |
| | C4BF6 | | 6 | 1.91 | 13.74 | |
| | C4BF8 | | 8 | 1.9 | 13.86 | |
| UU test | C4BF0 | 4 | 0 | 1.96 | 13.69 | 95 |
| | C4BF2 | | 2 | 1.94 | 13.70 | |
| | C4BF4 | | 4 | 1.93 | 13.69 | |
| | C4BF6 | | 6 | 1.91 | 13.74 | |
| | C4BF8 | | 8 | 1.9 | 13.86 | |

**Test materials**

Red sandstone soil Cement Basalt fibre

**Specimens preparation and curing**

Add water/Stir/Moisturise

Preparation of disintegration specimens

Preparation of UCS specimens

Preparation of UU specimens

Curing

**Test process**

Saturation

Disintegration test UCS test UU test

SEM test

**Fig 1. Test materials and test methods.**

water is included during mixing to achieve the optimum moisture content. The mixture is stirred thoroughly with a mixing tool every 30 minutes during the 2-hour delay. Finally, the mixture is layered into different molds within 1 hour.

The disintegration specimens were compacted into cubic specimens with a side length of 5 cm in three layers. For the UCS specimens, the mixed specimens were pressed into the test molds in three phases using a reaction frame and jack, forming cylindrical specimens with dimensions of φ100 mm × H100 mm. After 2 hours, the specimens were demolded using an electric demolding apparatus. The UU specimens were loaded into a triaxial specimen mold with dimensions of φ39.1 mm × H80 mm in five layers, and the cylindrical specimens were formed using the compaction method. All specimens were wrapped in plastic film immediately after demolding, and placed them to a constant temperature and humidity chamber for 7 days. The maintenance temperature was maintained at 20±2 °C, and the relative humidity was kept at ≥ 95%. After the maintenance period, these specimens were placed in a vacuum cylinder for pumping, continuing for 1 to 2 hours to achieve vacuum saturation once the vacuum level approached 1 atmosphere.

Disintegration test were conducted using a homemade water immersion disintegration test device. Initially, the net was immersed in a tank filled with a specific amount of water to zero the dynamometer. The net was then lifted, and the specimen was placed in the center. Both the net and specimen were smoothly immersed in the water to commence the test. During the disintegration process, the computer automatically recorded the dynamometer readings every second, while a camera captured images reflecting the condition of the specimen in the water. The disintegration test was deemed complete when the force gauge read 0 or remained stable. Each group consisted of two parallel specimens, and the disintegration rate was calculated using the following formula:

$$A_t = \frac{R_0 - R_t}{R_0} \times 100\%$$

(1)

In the equation: $A_t$ (%) represents the disintegration rate of the specimen at time $t$. $R_0$ (N) is the instantaneous stable reading of the force gauge at the beginning of the $R_t$ is the dynamometer reading at time $t$.

A microcomputer-controlled electronic universal testing machine was employed to conduct UCS tests on each group of specimens. The loading strain rate was set at 1.0 mm/min, with the tester capable of a maximum load of 20 kN. The axial load was applied through a strain control module, while the test data were automatically collected by a computer system. The UCS is defined as the peak strength observed on the axial stress-strain curve, with the final result determined as the average of three test sets. Furthermore, in order to examine the morphological characteristics of the specimens post-destruction, the test loading strains were controlled at approximately 5%.

Due to the high water stability of the cement cured soil, the triaxial compression test was conducted using a methodology distinct from that employed for conventional soft soil. In accordance with the recommendations of the Specification for mix proportion design of cement soil (JGJ/T 233−2011) [26], drainage and consolidation of cement cured soil is typically not required in general engineering. Therefore, this paper employs the undrained and unconsolidated shear (UU) test method. The TSZ-2 automatic triaxial instrument was employed to apply peripheral pressures of 50, 100, and 200 kPa, with a shear rate of 0.5 mm/min, while the data were automatically collected by a computer.

Some representative specimens were selected and mounted on the sample carrier plate using conductive adhesive, then vacuum-coated with gold using the GVC-1000 ion sputtering apparatus. Finally, the microstructure of these specimens was examined using the KYKY-EM6200 scanning electron microscope.

## 3. Test results and analyses

### 3.1. Disintegration test

Water is one of the most detrimental factors impacting the strength and stability of red sandstone roadbed fill. In order to ensure the stability of the roadbed, the fill material must possess strong water stability and high resistance to disintegration. The result of the disintegration test on different cement cured red sandstone soil are presented in Fig 2.

The cement dosage significantly influences the water stability of red sandstone soil. As shown in Fig 2, the addition of cement enhances the disintegration resistance of the specimens. With increasing cement content, the average disintegration rate decreases, leading to improved water stability. The disintegration rate of the C0BF0 and C3BF0 sample can be categorized into three distinct phases. Initially, both samples exhibited an exponential increase in disintegration rate. As time progressed, the rate of disintegration transitioned to a linear increase. Once the specimen reached a certain level of disintegration, the rate began to plateau. At this phase, the C0BF0 specimen had fully disintegrated, while the C3BF0 specimen remained stable with no further disintegration. In contrast, the disintegration rate of the C4BF0 and C5BF0 specimen increased rapidly at first, before gradually stabilizing.

As shown in Fig 3, the C0BF0 specimen fully disintegrated after 11 minutes of water immersion, demonstrating very poor water stability. However, after cement modification, the water stability of the specimens improved significantly. With a cement dosage of 3%, the disintegration rate of C3BF0 was approximately 24% after 20 minutes of immersion. When the cement dosage exceeded 4%, the specimen exhibited minimal disintegration. It suggests that the disintegration time of red sandstone soil increases with higher cement content, and disintegration is essentially eliminated once the cement dosage reaches a certain threshold. As shown in Fig 4, the improvement arises from the cement hydration products—primarily fibrous calcium silicate hydrate (C-S-H) and lamellar calcium hydroxide (C-H)—formed when cement reacts with red

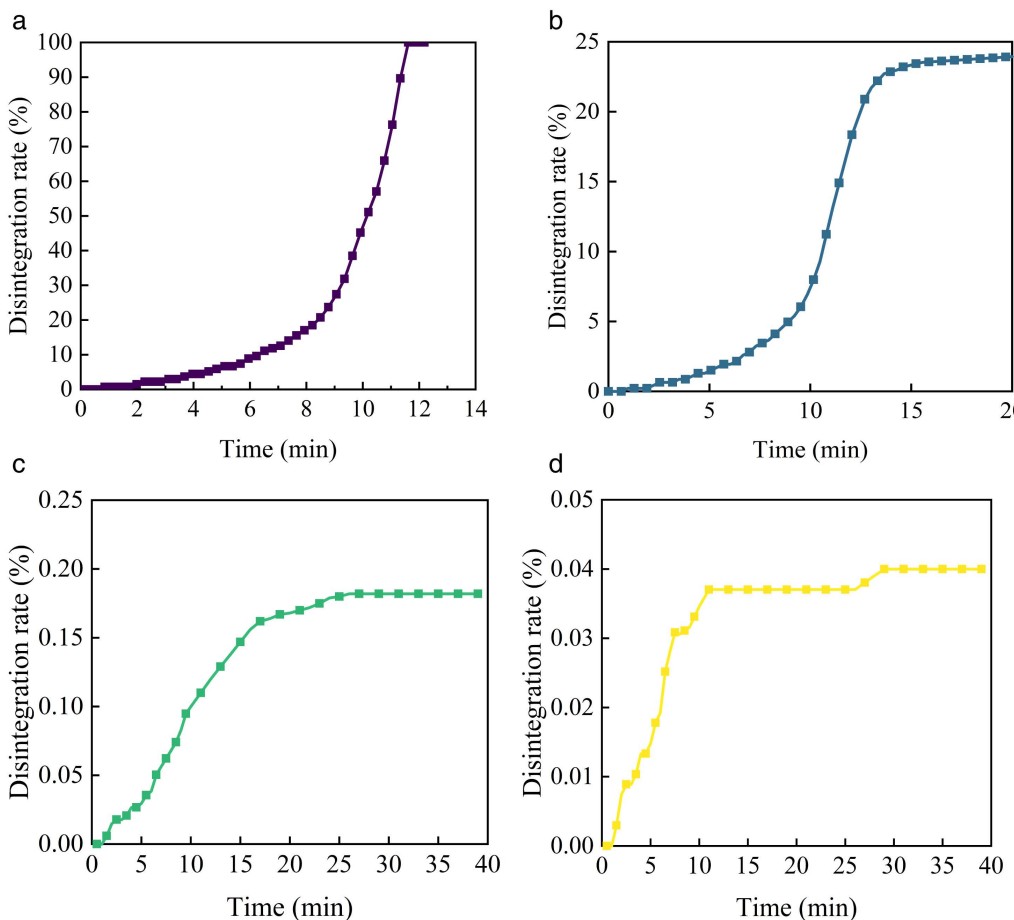

**Fig 2. Disintegration test curves of different cement cured red sandstone soil.** (a) C0BF0. (b) C3BF0. (c) C4BF0. (d) C5BF0.

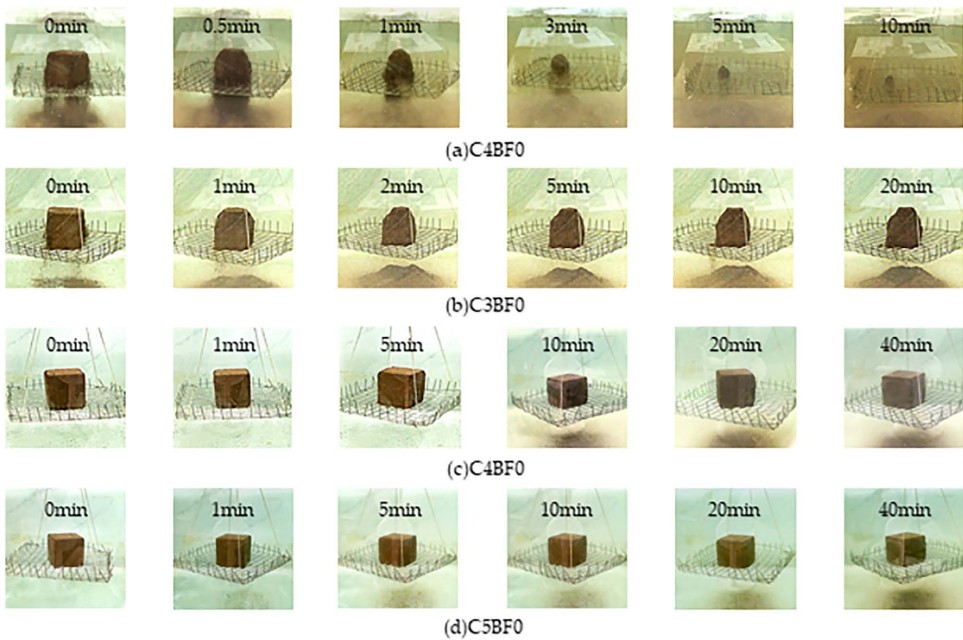

**Fig 3. Disintegration process of different cement cured red sandstone soil.**

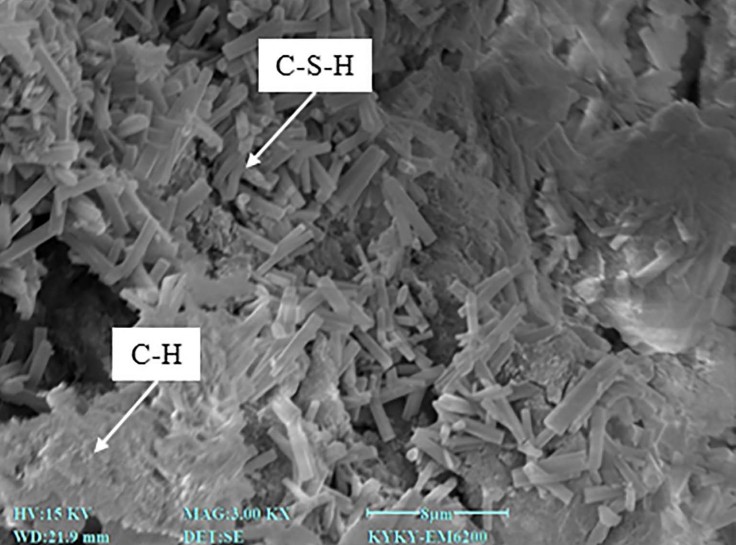

**Fig 4. Microstructure of cement cured red sandstone soil.**

sandstone soil. These hydration phases both fill the interparticle voids and establish a robust cementitious network that inhibits particle detachment, thereby markedly enhancing the soil's resistance to disintegration. Most research indicates that higher cement content as a stabilizer improves water stability [27,28]. However, increasing the cement dosage also raises material brittleness and construction costs. Considering these factors, a cement content of 4% is deemed optimal and will be used in subsequent tests.

## 3.2. Unconfined compressive strength (UCS) test

The compressive strength index of the soil is one of the important indicators of the physical and mechanical properties of the soil. Fig 5 demonstrates the effect of BF content on the UCS of cement cured red sandstone soil. As can be seen from the figure, the 7 d saturated UCS of all specimens under different compaction degrees is greater than 0.35 MPa, which meets the requirements of the specification of National Railway Administration of the People's Republic of China. Code for design of railway earth structure (TB 10001−2016) [29]. This result further supports the appropriateness of the selected 4% cement dosage.

BF contribute to further improving the compressive strength of the soil. As shown in Fig 5, the UCS of cement cured red sandstone soil initially increases and then decreases with the rise in BF content. For example, in the specimen with 95% compaction, the UCS of the C4BF0 specimen was 3.021 MPa. With the increase in BF content from 2‰ to 6‰, the UCS improved by 0.156 MPa, 0.378 MPa, and 0.765 MPa, representing increments of 5.18%, 12.55%, and 25.40%, respectively. However, when the BF content increased to 8‰, the UCS of the C4BF8 specimen was lower than that of C4BF6 specimen. This indicates that there is an optimal dosage of BF for enhancing the UCS of cement cured red sandstone soil. Once the optimal BF dosage is exceeded, the strengthening effect diminishes. This can be attributed to the excellent tensile properties of BF. An appropriate amount of BF can create a spatial reinforcement structure within the cement cured red sandstone soil, enhancing the overall performance of soil. This structure effectively inhibits the generation and development of internal cracks under axial loading, thereby improving the UCS of the cement cured red sandstone soil. However, as the fiber dosage increases, the BF may become unevenly distributed within soil, weakening the reinforcing effect. Consequently, the UCS of the cement cured red sandstone soil decreases at higher fiber dosage.

The degree of compaction significantly influences the compressive properties of the soil. As illustrated in Fig 5, for a given amount of BF, the UCS of the specimens gradually increases with the degree of compaction. Notably, the rate of strength increase is greater at higher compaction levels. The average strength growth rates of the improved soil were 11.35%, 17.41%, and 39.91% when the compaction degree was raised from 90% to 92%, 93%, and 95%, respectively. This is because a higher degree of compaction results in denser soil, which increases the contact area of soil particle. Consequently, this enhances the cementation and embedding effects, leading to higher UCS.

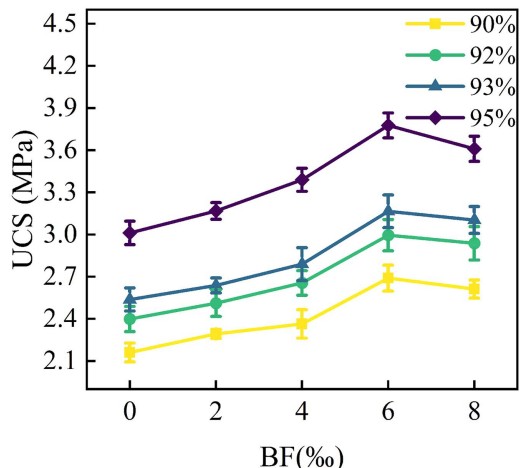

**Fig 5. Effect of BF dosage on the UCS of cement cured red sandstone soil.**

BF significantly enhance the damage characteristics of cement cured red sandstone soil under axial deformation. The morphology of different specimens following damage in the UCS test is illustrated in Fig 6. It is evident that the damage morphology of the specimens exhibits split tensile failure. A certain thickness of material on the radial surface of the specimens ruptured into several blocks along the radial direction, yet remained connected to the main body, maintaining a single unit structure. Overall, the specimens retain a cylindrical appearance. During the loading of specimens, small macroscopic cracks initially appeared in the top surface region. As the loading increased, these cracks developed slowly, extending both inward and downward, resulting in several vertical cracks of varying lengths. Eventually, the cracks interconnected, creating a continuous network throughout the specimen. The C4BF0 specimen was damaged with a vertical crack running the entire length of the specimen with a considerable crack width. After the addition of BF, the width of the cracks formed during specimen damage decreased. Notably, the C4BF6 specimen displayed a dense, roughly parallel cluster of cracks with significantly shorter lengths, and the axial strain generated during damage was slightly greater than that of the specimens with the other three fiber dosages. This illustrates that deformation damage of cement cured red sandstone soil is primarily governed by cement in the final damage mode, while fiber predominantly influence the extent of crack propagation.

### 3.3. Undrained and unconsolidated shear (UU) test

Soil in practical engineering is subjected to various loads, including perimeter pressure, axial load, and dynamic load. In order to simulate the actual stress state of soils under these different loads, assess their resistance to deformation and failure modes, UU tests were conducted on various improved soil. The deviatoric stress-axial strain curves from tests are presented in Fig 7. The trends of the deviatoric stress-axial strain curves for cement cured red sandstone soil exhibit homogeneity across different BF contents, as illustrated in the Fig 7. Each curve demonstrates that, with a constant perimeter pressure, the deviatoric stress increases rapidly to its peak value as axial strain increases, before gradually decreasing to the residual strength, thereby exhibiting clear strain softening characteristics. This phenomenon can be attributed to the dense structure of the red sandstone soil modified by cement. During the soil shear process, greater energy is required to overcome soil particle interlocking, resulting in reduced swelling and increased strength. However, as strain develops and the rearrangement of soil particle occurs, the strength diminishes, which is indicative of strain softening. As illustrated in Fig 7, the deviatoric stress-axial strain curve can be broadly categorized into four distinct phases:

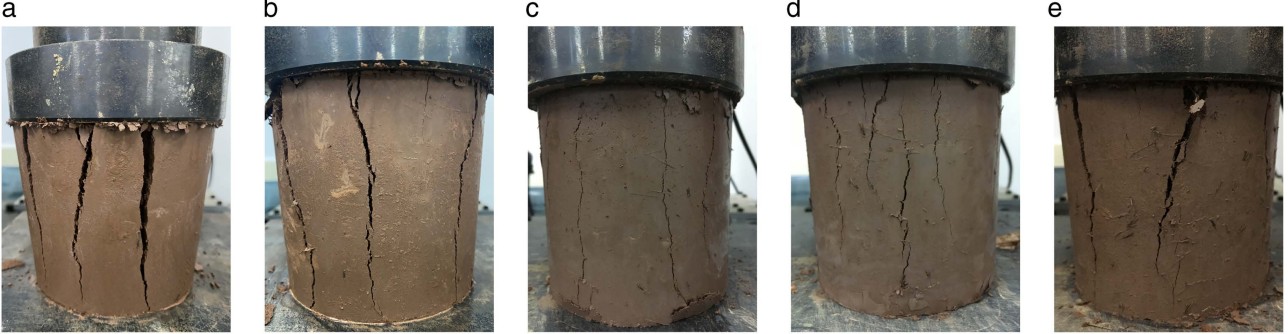

**Fig 6. Morphology of different specimens after damage from UCS tests.** (a) C4BF0. (b) C4BF2. (c) C4BF4. (d) C4BF6. (e) C4BF8.



**Fig 7. Deviatoric stress-axial strain curve of cement cured red sandstone soil with different BF dosages.** (a) C4BF0. (b) C4BF2. (c) C4BF4. (d) C4BF6. (e) C4BF8.

(a) Densification Phase: The deviatoric stress-axial strain curve exhibits a concave shape due to the closure of fine cracks within the geotechnical medium. In this phase, both the deviatoric stress and axial strain remain low, resulting in only a slight reduction in the specimen's volume.

(b) Elastic Deformation Phase: In this phase, the deviatoric stress-axial strain curve closely resembles a straight line, with deviatoric stress increasing rapidly as axial strain increases, demonstrating linear elastic behavior.

(c) Plastic Deformation Phase: During this phase, the deviatoric stress-axial strain curve gradually elongates while the deviatoric stress increases at a slower rate. As the deviatoric stress reaches a certain threshold, the specimen's volume begins to expand, leading to the continuous development of cracks. When damage accumulates to a certain extent, specimen failure occurs, coinciding with the maximum deviatoric stress.

(d) Residual Damage Phase: When the axial strain surpasses the point corresponding to peak deviatoric stress, the deviatoric stress on the specimen decreases significantly as axial strain increases, eventually stabilizing. During this phase, cracks within the specimen develop rapidly, gradually forming a shear surface. At this point, the specimen retains a certain level of residual strength.

BF improved the triaxial shear properties of cement cured red sandstone soil. As illustrated in Fig 7(b) to (e), the general shape of the deviatoric stress-axial strain curve for the cement cured red sandstone soil remained unchanged with the addition of BF, though variations were observed at different phases of deformation. The differences in the deviatoric stress-axial strain curves among specimens with varying reinforcement levels are minimal during the densification and elastic deformation phase. This may be attributed to the limited deformation in the initial phase, where the BF primarily contribute to controlling larger deformations and crack development [30]. In the plastic deformation phase, the differences between the deviatoric stress-axial strain curves of cement cured red sandstone soil and BF reinforced cement cured red sandstone soil gradually become apparent. Specifically, the plastic deformation phase of the cement cured red sandstone soil is shorter, and the deviatoric stress increases more rapidly, while the BF reinforced cement cured red sandstone soil exhibits a longer plastic deformation phase and a higher peak axial stress. This discrepancy arises because the cement cured red sandstone soil lacks sufficient toughness during fracture propagation, leading to rapid stress growth and an early failure point. In contrast, the BF create a bridging structure between soil particle, enhancing cohesion and friction, delaying crack initiation and propagation, and enabling the soil to maintain a high load-bearing capacity over a larger strain range. This extends the plastic deformation phase and increases peak strength. During the residual damage phase, the residual strength of BF reinforced cement cured red sandstone soil is significantly higher compared to cement cured red sandstone soil, demonstrating enhanced toughness. This is because BF embedded within the cracks provides additional tensile and shear resistance when the crack in the soil propagates. As the soil undergoes further deformation, it must overcome the resistance offered by BF, thereby increasing its residual strength.

The effect of varying BF dosage on the peak deviatoric stress of cement cured red sandstone soil is illustrated in Fig 8. As shown in the figure, under constant circumferential pressure, the peak deviatoric stress of the soil increases as the BF content rises from 0‰ to 6‰. At a dosage of 6‰, the BF have the most significant impact, enhancing the peak deviatoric stress of the cement cured red sandstone soil by 31.13% to 39.48%. This is because the cement cured red sandstone soil specimen primarily relies on the skeleton formed by cemented soil particle, which are bonded by the hydration products of the cement. In contrast, the specimen with 6‰ BF content forms a "fibre network" among the soil particle through the random distribution and interconnection of the BF, effectively limiting the lateral deformation of the soil. Additionally, the cement hydration products increase the adhesion between the BF and soil particle, significantly enhancing the shear performance. The peak deviatoric stress decreased when the BF dosage was increased to 8‰. This phenomenon indicating that more BF is not always beneficial. It could be because excessive BF become cross-stacked within the soil, hindering the bonding between soil particle and the cement hydration products, which in turn

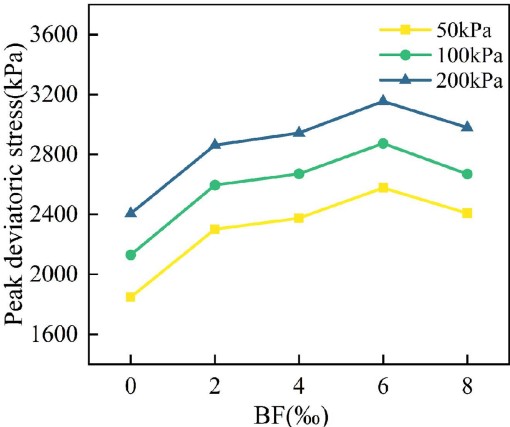

**Fig 8. Effect of BF dosage on the peak deviatoric stress of cement cured red sandstone soil.**

reduces the peak deviatoric stress. It has been demonstrated in previous studies that there is a critical fibre dosage that optimizes soil strength [31,32]. When this threshold is exceeded, the strength of the soil begins to decrease, which aligns with the findings of this paper.

When the BF dosage remains constant, the peak deviatoric stress increases as the circumferential pressure rises. This is because circumferential pressure acts as a radial constraint on the specimen, limiting its lateral deformation. As the axial stress increases, the circumferential pressure effectively enhances the specimen's shear resistance and prevents slip damage along the shear surface, thereby improving the specimen's load-bearing capacity.

The effect of different BF dosages on the cohesion and internal friction angle of cement cured red sandstone soil is illustrated in Fig 9. It can be observed that as the amount of BF increases, both cohesion and internal friction angle initially increase and then decrease. However, the magnitude of change for the two properties varies. Specifically, the internal friction angle is less affected by BF dosage. When the BF dosage was varied in the range of 0‰~6‰, the internal friction angle increased linearly with the BF dosage, and started to decrease after reaching a peak value of 40.96°. At the optimum BF dosage of 6‰, the internal friction angle is only enhanced by 1.49% compared to the C4BF0 specimen. On the contrary, the incorporation of BF has a more pronounced effect on cohesion. At low BF dosages, the addition of BF significantly enhances soil cohesion. This improvement occurs because BF can be uniformly distributed within the soil, enhancing the interlocking and friction between soil particle and dispersing stress through a "bridge effect," thereby substantially

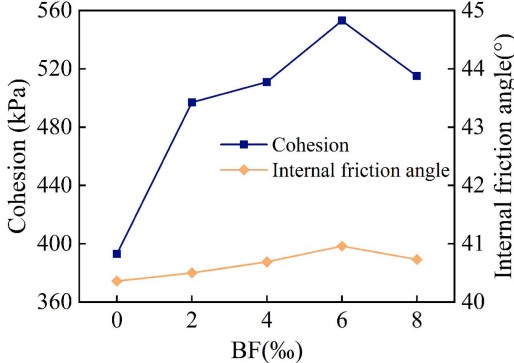

**Fig 9. Effect of BF dosage on cohesion and internal friction angle of cement cured red sandstone soil.**

increasing the soil's cohesion. As the BF dosage increases, cohesion continues to rise. At a BF dosage of 6‰, the cohesion reaches 552.93 kPa, which is 40.74% higher than that of the C4BF0 specimens. However, when the BF content is excessively high, the BF may aggregate and result in uneven distribution. This disrupts the close contact between soil particle, creating additional pores and weakly connected areas. Consequently, the bonding effect between the BF and soil particle is weakened, leading to a decrease in cohesion. This result indicated that the variation in triaxial peak deviatoric stress of cement cured red sandstone soil with different BF dosages was primarily attributed to cohesion rather than the internal friction angle, a phenomenon that aligns with the findings reported by Han et al [33].

In order to characterize the deformation and stability of the specimens, an approximate straight line segment of the deviatoric stress-axial strain curve was selected for fitting, starting from the point where the strain reaches 1% and extending to the peak strength point. The slope of this straight line represents the modulus of elasticity, which describes the deformation characteristics of the soil under conditions of low stress and minimal deformation. This modulus is crucial for ensuring the stability and safety of the roadbed during the early phases of construction and train operation [34].

Fig 10 illustrates the effect of different BF dosages on the elastic modulus of cement cured red sandstone soil. As shown in the figure, the elastic modulus for red sandstone soil mixed with cement and BF exceeds 200 MPa, indicating that the soil exhibits a high bearing capacity. This characteristic allows for minimal deformation and settlement under train loads, thereby maintaining track level and stability, and ensuring the comfort and safety of train operations. Under the same circumferential pressure, the elastic modulus of cement cured red sandstone soil specimens exhibited a trend of initially increasing and then decreasing with increasing BF dosage. For instance, at a circumferential pressure of 50 kPa, the elasticity modulus increased by 6.93%, 9.76%, and 13.04% compared to the C4BF0 specimen for BF dosages of 2‰, 4‰, and 6‰, respectively. However, when the BF dosage reached 8‰, the elastic modulus decreased by 4.08 MPa compared to the C4BF6 specimen. This phenomenon can be attributed to the excessive accumulation of BF, which results in increased porosity and a weakening of the interfacial interaction between the BF and the soil particle at high BF content. At the same BF admixture, increasing the confining pressure results in a higher elastic modulus for the soil. This is primarily because the confining pressure enhances the friction and interlocking between soil particle, reduces the presence of pores and microcracks, improves the contact network among soil particle, and increases the shear strength of the soil. Consequently, the soil exhibits greater stiffness and a higher modulus of elasticity when subjected to external forces.

In this section, the strain corresponding to the peak stress of a cured soil specimen is defined as the failure strain. It indicates the maximum amount of deformation the material can endure before failure. Fig 11 illustrates the failure strain

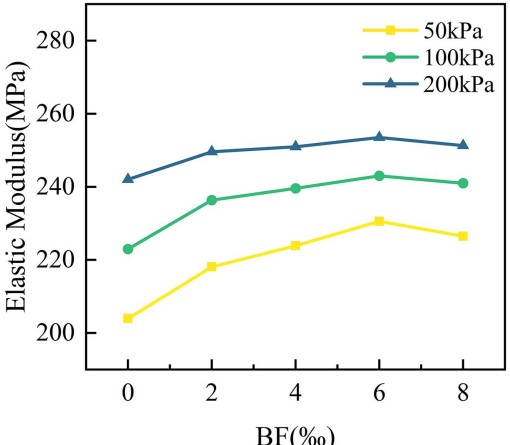

**Fig 10. Effect of BF dosage on the modulus of elasticity of cement cured red sandstone soil.**

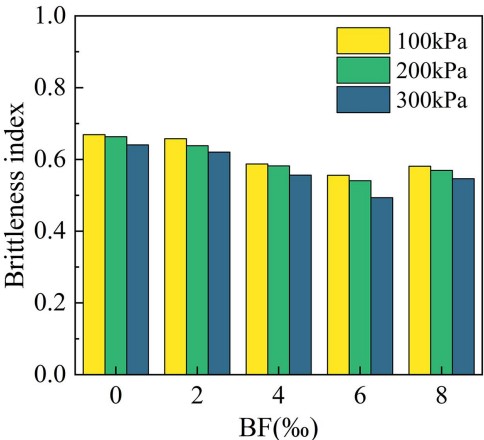

**Fig 11. Effect of BF dosage on the failure strain of cement cured red sandstone soil.**

of cement cured red sandstone soil under varying levels of perimeter pressure and BF reinforcement. The failure strains of specimens with BF incorporation are higher than those without BF at various enclosure pressures, as shown in Fig 11. It indicates that BF incorporation enhances the deformation resistance of cement cured red sandstone soil. As the BF admixture increased from 0‰ to 6‰, the failure strain exhibited an increasing trend, suggesting that increasing BF dosage within this range can improve the ductility and structural stability of the soil, allowing it to withstand greater deformation before failure. However, when the BF dosage reached 8‰, the failure strain decreased, indicating a reduction in ductility. This may be attributed to the reinforcing effect of the BF approaching saturation at this dosage. When the BF dosage is excessive, agglomeration can occur, weakening the interfacial bonding between BF and soil particle in certain areas. This leads to a decrease in the overall deformation capacity and bearing capacity of the soil under stress. Therefore, it is essential to control the BF dosage in soil improvement to maximize the enhancement effects and avoid the negative consequences associated with an overly high dosage. The failure strain of the specimen increases with higher perimeter pressure at the same BF doping level. This effect is primarily attributed to the enhanced confinement of the BF within the soil matrix under elevated perimeter pressure, allowing the BF to provide greater tensile and shear resistance during damage process. Consequently, this resistance slows down the damage process, enabling the soil to withstand greater deformation, which results in an increase in failure strain.

Cement enhances the strength of the soil but also increases its brittleness, which can lead to subsidence or structural damage to the roadbed in the event of sudden destabilization. This poses a potential risk for railway foundations. In order to quantify the effect of BF on the toughness of cement cured red sandstone soil, and assess the performance of BF reinforced cement cured red sandstone soil under extreme loads, this section adopts a quantitative evaluation method for rock brittleness characteristics based on the full stress-strain curve [35]. The brittleness index $I_B$ is defined as the relationship between soil brittleness and the post-peak stress drop, taking into account both peak and residual strength:

$$I_B = \frac{\tau_p - \tau_r}{\tau_p}$$

(2)

In the equation: $\tau_p$ (kPa) represents the peak deviatoric stress. $\tau_r$ (kPa) denotes the residual deviatoric stress, and $I_B$ indicates the magnitude of the post-peak stress drop. A larger value of $I_B$ signifies greater brittleness of the soil.

Fig 12 illustrates the brittleness index of cement cured red sandstone soil at various levels of perimeter pressure and BF reinforcement. The data indicate that the incorporation of BF can effectively reduce the brittleness of cement cured red

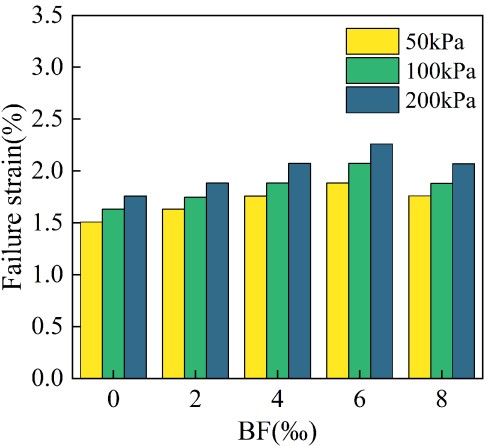

**Fig 12. Effect of BF dosage on the destructive strain of cement-cured soil.**

sandstone soil to a certain extent. Notably, the brittleness decreases gradually with increasing BF dosage before experiencing a slight increase. The enhancement of soil brittleness due to BF is most pronounced at an optimal BF dosage of 6‰. This test result contrasts with the destructive strain, reinforcing the viewpoint proposed by V. Hucka and B. Das that "the lower the strain at destruction, the greater the degree of brittleness" [36]. The brittleness index decreases as the peripheral pressure increases for a given amount of BF doping, aligning with the established effect of peripheral pressure on rock brittleness.

## 4 Discussion

### 4.1. Mechanism of BF reinforced cement cured red sandstone soil

In order to analyze the mechanism of BF reinforced cement cured red sandstone soil, a typical reinforced specimen, C4BF6, was selected for SEM testing. The SEM image is presented in Fig 13. Fig 13 (a) clearly illustrate the distribution of BF in cement cured red sandstone soil, showing them either as individual BF or in clusters. Scattered BF act as individual reinforcements, while overlapping BF create a three-dimensional reinforcement network within the soil. Observation of the bonding state of the BF in locally magnified images revealed that the outer surface of the BF was tightly bonded to the

a
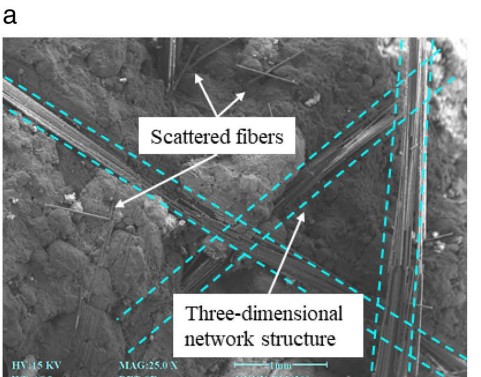

b
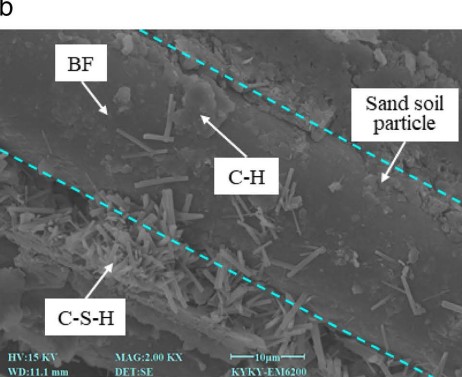

**Fig 13. SEM image of C4BF6 specimen.**

hydration products (C-H and C-S-H) and soil, forming a well-bonded interface, as shown in Fig 13 (b). Additionally, a small amount of hydration products was adsorbed onto the BF surface, demonstrating significant cementation capacity.

The mechanism by which cement enhances the strength of soil primarily involves the cementation and filling effects of cement hydrates [37,38]. Through hydration reactions, cement generates hydration products that serve two key functions: first, they fill the pores between soil particle, improving the soil's compactness; second, they act as a binding agent between soil particle. This adhesive force enables the originally loose sand and soil particle to bond closely together, significantly enhancing the overall strength and stability of soil. The role of BF in enhancing the strength of soil primarily depends on the interfacial friction between the BF surface and soil [39]. This interfacial friction enables BF to act as a bridging agent during crack formation and expansion. By transferring and dispersing stress, BF help to prevent crack propagation, thereby improving the strength and stability of soil. BF reinforced cement cured red sandstone soil is a complex multiphase material, and the interfacial contact interactions among the BF, soil, and hydration products primarily involve friction, occlusion, and cementation [40]. The concept of this specific contact mechanism is illustrated in Fig 14. As shown in the figure, the hydration products of cement bond the BF to the surrounding soil particle, forming an integrated stress structure. Cement hydration enhances the cementation and frictional occlusion at the BF-soil interface. When a potential sliding surface develops within the soil, the BF and hydration products at this interface experience simultaneous stress. However, due to the differences in material properties among these components, various deformations occur, leading to misalignment. At this point, the stress along the potential slip surface is transmitted through the BF to the soil and hydration products. This tensile stress in the BF spreads to the surrounding soil until the BF are either pulled off or extracted, thereby facilitating load sharing and improving the soil's ductility.

The distribution of BF in the soil at different fiber dosages is illustrated in Fig 15. It is evident that BF is randomly dispersed within the soil, forming a spatially reinforced structure. At low fiber dosages, the BF is scattered throughout the specimen, relying primarily on individual fiber for one-dimensional tensile reinforcement. The reinforcing effect is largely dependent on the tensile strength of the BF, the frictional resistance at the interface between the BF and the sand particle, and the magnitude of the adhesive force, resulting in a limited enhancement. With increasing fiber doping, BF overlap and intertwine to form a three-dimensional force network, significantly enhancing the overall performance of the soil and improving its ability to resist deformation and damage. However, when the fiber dosage exceeds 6‰, the static electricity carried by the BF can lead to mutual adsorption, resulting in agglomeration. This phenomenon creates weak planes within the soil, diminishing the reinforcing effect. Consequently, high fiber dosages may lead to a reduction in the hydraulic soil's resistance to damage and deformation, aligning with the results from the UCS and UU tests.

## 4.2. Destruction mechanism of reinforced cement cured red sandstone soil

Fig 16 illustrates the damage process of BF reinforced cement cured red sandstone soil. During the initial loading phase of the specimen, the sand particle, hydration products, and BF gradually compact, resulting in small displacements.

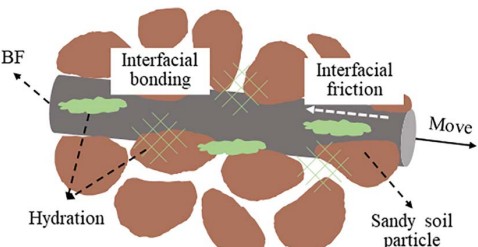

**Fig 14. Conceptual model of BF-hydration product-sand particle interfacial interaction.**

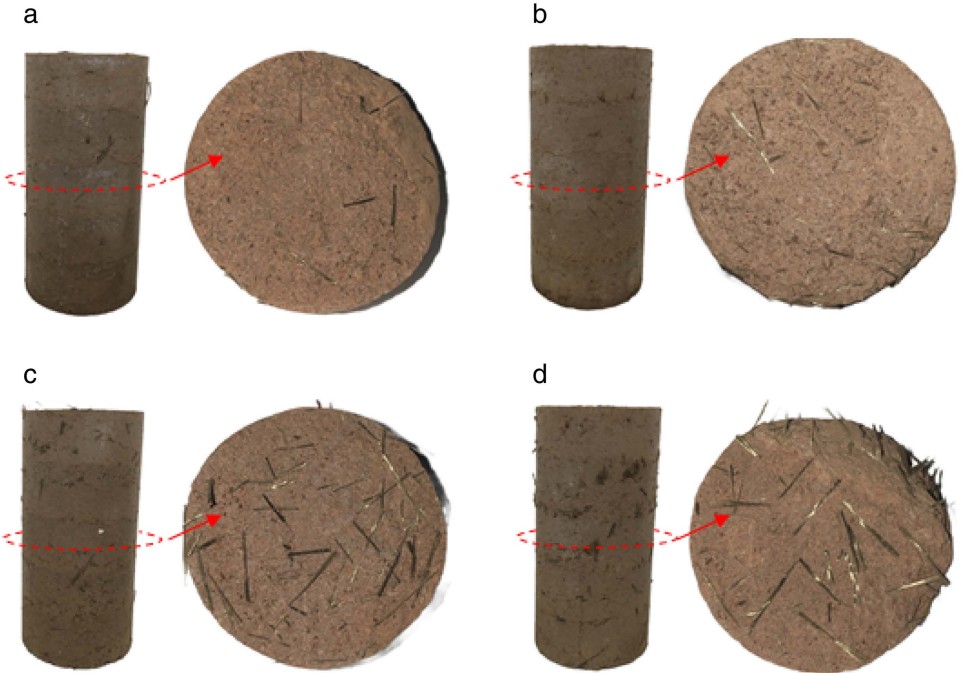

**Fig 15. The distribution of BF in the soil at different fiber dosages.** (a) C4BF2. (b) C4BF4. (c) C4BF6. (d) C4BF8.

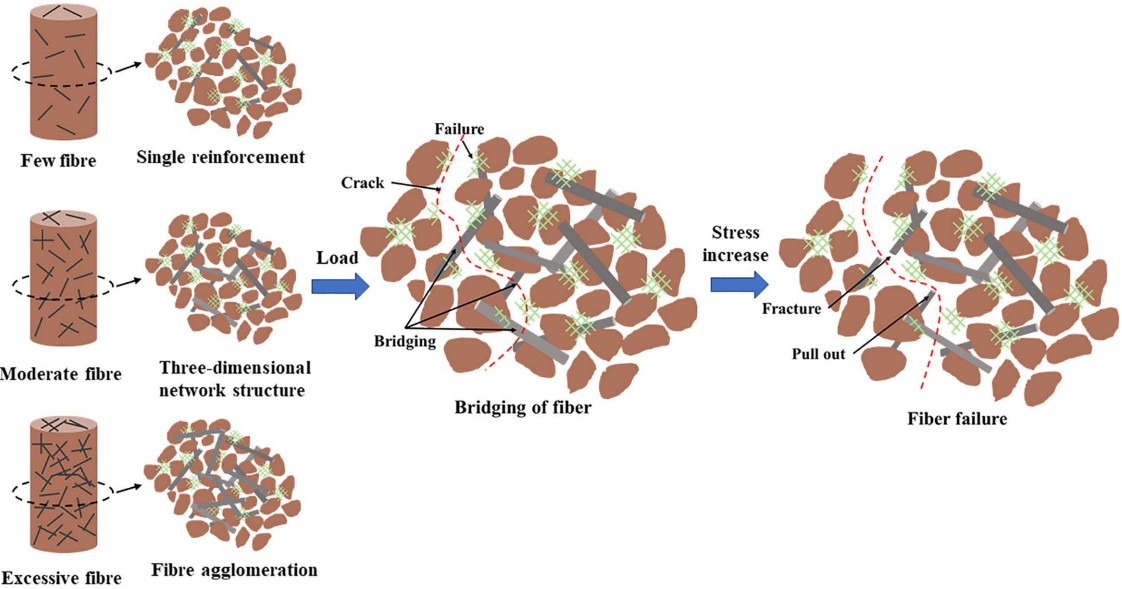

**Fig 16. Damage process of BF reinforced cement cured red sandstone soil.**

With further increases in stress, the hydration products cannot withstand the excessive pressure and begin to rupture, losing their ability to effectively cement the sand particle. Consequently, internal cracks within the soil continue to form and expand. At this phase, the BF at the crack act as a "bridge," preventing further damage to the specimen through the

adhesion between BF and hydration products on either side of the crack, as well as the friction between BF and soil interface. As the degree of damage increases, the BF gradually elongate and thin, eventually slipping, pulling out, or breaking while attempting to pull sand particle. This results in weakened or failed BF reinforcement, leading to the extension of microcracks and the formation of through cracks, ultimately causing damage to the specimen. When the fiber content is low, damage to the soil primarily results from the degradation of hydration products and the weakening of cohesion among sand particle, leading to the formation and penetration of cracks. As the fiber content increases, the failure mode of the soil increasingly involves BF pullout and its influence on crack development.

## 5 Conclusion

In this study, the physico-mechanical properties and improvement effects of cement composite BF reinforced red sandstone soil were investigated by Disintegration tests, UCS tests and UU tests. Additionally, the mechanisms of BF reinforcement and damage in cement cured red sandstone soil were analyzed through SEM testing. The following conclusions were drawn:

(a). The water stability of red sandstone soil under submerged conditions is notably inadequate. However, the incorporation of cement significantly reduces the disintegration rate. As the cement content increases, the disintegration resistance of the specimens improves substantially, leading to a gradual decrease in the average disintegration rate and a marked enhancement in water stability. When the cement content reaches 4%, the specimens exhibit negligible disintegration. Therefore, in order to ensure the stability of red sandstone roadbed in a water environment, it is advisable to maintain cement dosing at 4%.

(b). Basalt fiber (BF) enhances the compressive properties of cement cured red sandstone soil. As the fiber dosage increases, the UCS of specimens at various compaction levels initially rises and then declines. At 95% compaction, the maximum UCS is attained with a BF dosage of 6‰, resulting in a 25.40% improvement compared to cement cured red sandstone soil without BF.

(c). The incorporation of BF enhances the shear properties of cement cured red sandstone soils. As fiber dosage increases, the deviatoric stress-axial strain curve initially rises and then falls. Within the range of 0‰ to 6‰ fiber doping, peak deviatoric stress and cohesion are positively correlated with fiber content, attributable to the individual reinforcing effects of BF and the development of a three-dimensional reinforcement network. However, when the fiber dosage exceeds 6‰, uneven distribution can occur within the soil, creating weak cross-sections that diminish the reinforcing effect of the BF, ultimately leading to a reduction in shear and cohesive strength. At the optimal fiber dosage of 6‰, peak deviatoric stress improved by 31.13% to 39.48%, while cohesion increased by 40.74%. In contrast, the internal friction angle experienced only a marginal increase of 1.49%.

(d). BF significantly enhanced the deformation and stability properties of cement cured red sandstone soil. With the addition of BF, the elastic modulus and peak failure strain of the cement cured soil initially increased and then declined. The specimens exhibited markedly reduced strain softening and brittleness. Under a confining pressure of 100 kPa, the BF content of 6 ‰ enhanced the peak failure strain by 26.92% and decreased the brittleness index by 18.48%. In summary, for the modified red sandstone roadbed filler used in this paper, the optimum ratio of red sandstone, water, cement and BF is 1:0.14:0.04:0.006. For red sandstone from other sources with varying composition or physical properties, the water–cement ratio and BF dosage should be tailored via small-scale laboratory tests to its specific mineralogy and gradation.

## Author contributions

**Funding acquisition:** Yao Long, Qi-yun Wang.

**Resources:** Yao Long.



**Supervision:** Jun-hua Chen.

**Writing – original draft:** Jie-jie Mu.

**Writing – review & editing:** Jie-jie Mu.

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
