## [Decision Letter · Decision Letter 0]

22 Jun 2025

Dear Dr. Chen,

Thank you for submitting your manuscript to PLOS ONE. After careful consideration, we feel that it has merit but does not fully meet PLOS ONE’s publication criteria as it currently stands. Therefore, we invite you to submit a revised version of the manuscript that addresses the points raised during the review process.

We look forward to receiving your revised manuscript.

Kind regards,

Makungu Marco Madirisha

Academic Editor

PLOS ONE

Journal Requirements:

“(1) Hunan Natural Science Foundation Sectoral Joint Fund (2024JJ8021).

(2) Hunan Provincial Department of Education Scientific Research Project (22B0958).

(3) Guangxi Natural Science Foundation Project (2022GXNSFAA035485).

(4) Science and Technol-ogy Base and Talent Special Project (GUIKE AD21220051).

(5) Scientific Research Pro-ject of the Hunan Provincial Department of Education, China (21B0803).”

Please state what role the funders took in the study. If the funders had no role, please state: 'The funders had no role in study design, data collection and analysis, decision to publish, or preparation of the manuscript.'

6. PLOS requires an ORCID iD for the corresponding author in Editorial Manager on papers submitted after December 6th, 2016. Please ensure that you have an ORCID iD and that it is validated in Editorial Manager. To do this, go to ‘Update my Information’ (in the upper left-hand corner of the main menu), and click on the Fetch/Validate link next to the ORCID field. This will take you to the ORCID site and allow you to create a new iD or authenticate a pre-existing iD in Editorial Manager.

Reviewers' comments:

Reviewer's Responses to Questions

**Comments to the Author**

1. Is the manuscript technically sound, and do the data support the conclusions?

Reviewer #1: Yes

Reviewer #2: Yes

Reviewer #3: Partly

Reviewer #4: Yes

2. Has the statistical analysis been performed appropriately and rigorously?

Reviewer #1: Yes

Reviewer #2: Yes

Reviewer #3: Yes

Reviewer #4: No

3. Have the authors made all data underlying the findings in their manuscript fully available?

Reviewer #1: Yes

Reviewer #2: Yes

Reviewer #3: Yes

Reviewer #4: Yes

4. Is the manuscript presented in an intelligible fashion and written in standard English?

Reviewer #1: Yes

Reviewer #2: Yes

Reviewer #3: No

Reviewer #4: Yes

Reviewer #1: The manuscript analyses the effect of basalt fiber on cement cured red sandstone soil mechanical properties. Experimental data are utilized in the research for achieving a proper result. It is important to note that the manuscript has a good structure and the subject is a applicable one so that be published in the journal.

Reviewer #2: There is not any especial comments unless some figures are small and they would be more suitable if presented in larger sizes. Of course , there is some limitations for the paper pages. Additionally, it would be more practical if the results were applied to a true project.

Reviewer #3: Comments

1. Tables 1, 2, and 3 is about the physical properties of red sand stone soil. However, it is more important to related the result with the standards. As well as, the chemical properties, composition needs to be included

2. How the dosage of cement and red sand stone soil is selected? Have you done pilot study?

3. What are the main performance advantages of short-cut basalt fiber (BF) compared to plant fibers in construction applications? Please clarify it

4. Disintegration test is Limited in explanation and mechanisms. Please explain it well by explain the mechanisms behind how cement improves disintegration resistance.

5. How much do hydration products contribute to the strength gain in quantitative terms? Is there data or reference that quantifies the improvement in strength due to cement hydration?

6. What is the exact nature of the bond between basalt fibers and hydration products?

7. What is the effect of fiber dosage and length on the stress transfer efficiency?

Reviewer #4: The manuscript looks good but there are few comments to be addressed before it can be accepted for publication consideration.The manuscript investigates the Study on Static Properties and Mechanism of Basalt Fibre Reinforced Cement Cured Red Sandstone Soil, aiming to determine the optimal mix design for improving roadbed filling material in areas rich in red sandstone.

The topic is highly relevant for sustainable engineering and infrastructure, particularly in regions where red sandstone is abundant. The use of basalt fiber, a relatively eco-friendly material, aligns with modern sustainability goals. However, the manuscript requires some improvements in terms of scientific depth, clarity of results, and discussion coherence.

**Do you want your identity to be public for this peer review?** For information about this choice, including consent withdrawal, please see our Privacy Policy

Reviewer #1: No

Reviewer #2: **Yes: ** Arash Barjasteh

Reviewer #3: No

Reviewer #4: No

---

## [Author Response · Author response to Decision Letter 1]

27 Jul 2025

1.Re: agree with the changes. Additional clarification has been made in Conclusion (d).

2.Re: agree with the changes. It has been indicated in the Conclusion (d) how the improved technology can be applied to practical works.

3.Re: agree with the changes. The full text has been carefully proofread and grammatical issues have been corrected.

4.Re: agree with the changes. The mechanism of cement influence has been explained in further depth in Test results and analyses - Disintegration test.

5.Re: it is clear that the brittleness of different soils increases with the addition of cement, as has been illustrated in references 26-27. In addition, studies have been carried out on the causes of increased soil brittleness by cement as well as its manifestations, which are not the focus of this paper.

6.Re: a more in-depth explanation of the effect of BF content on ucs, and how BF alters the load transfer mechanism, interfacial bonding and structural integrity is in the Discussion section of the paper.

7.Re: it's already been stated in the paper that it's excessive BF content.

8.Re: agree with the change. Specific values have been added to the conclusions for clarification.

9.Re: agree with the change. Conclusion (d) has been given to give the mixing ratios that can be used in practice in specific projects.

---

## [Decision Letter · Decision Letter 1]

26 Sep 2025

Study on Static Properties and Mechanism of Basalt Fibre Reinforced Cement Cured Red Sandstone Soil

PONE-D-25-07153R1

Dear Dr. Chen,

We’re pleased to inform you that your manuscript has been judged scientifically suitable for publication and will be formally accepted for publication once it meets all outstanding technical requirements.

Kind regards,

Jiaolong Ren

Academic Editor

PLOS ONE

Additional Editor Comments (optional):

Reviewers' comments:

Reviewer's Responses to Questions

**Comments to the Author**

Reviewer #1: All comments have been addressed

2. Is the manuscript technically sound, and do the data support the conclusions?

Reviewer #1: Yes

3. Has the statistical analysis been performed appropriately and rigorously?

Reviewer #1: Yes

4. Have the authors made all data underlying the findings in their manuscript fully available?

Reviewer #1: Yes

5. Is the manuscript presented in an intelligible fashion and written in standard English?

Reviewer #1: Yes

Reviewer #1: (No Response)

**Do you want your identity to be public for this peer review?** For information about this choice, including consent withdrawal, please see our Privacy Policy

Reviewer #1: No

---

## [Editor Report · Acceptance letter]

PONE-D-25-07153R1

PLOS ONE

Dear Dr. Chen,

I'm pleased to inform you that your manuscript has been deemed suitable for publication in PLOS ONE. Congratulations! Your manuscript is now being handed over to our production team.

Kind regards,

on behalf of

Dr. Jiaolong Ren

Academic Editor

PLOS ONE